# Integration of biological data via NMF for identification of human disease-associated gene modules through multi-label classification

**Syed Alberuni** [1], **Sumanta Ray** [1,2] *

1 Department of Computer Science & Engineering, Aliah University, Kolkata, West Bengal, India, 2 Data Science Unit, The West Bengal National University of Juridical Sciences, Kolkata, West Bengal, India

☯ All these authors are contributed equally to this work.

* sumanta.ray@aliah.ac.in

**Data Availability Statement:** All relevant data are within the manuscript and its Supporting information files.

## Abstract

Proteins associated with multiple diseases often interact, forming disease modules that are critical for understanding disease mechanisms. This study integrates protein-protein interactions (PPIs) and Gene Ontology data using non-negative matrix factorization (NMF) to identify gene modules associated with human diseases. We leverage two biological sources of information, protein-protein interactions (PPIs) and Gene Ontology data, to find connections between novel genes and diseases. The data sources are first converted into networks, which are then clustered to obtain modules. Two types of modules are then integrated through an NMF-based technique to obtain a set of meta-modules that preserve the essential characteristics of interaction patterns and functional similarity information among the proteins/genes. Each meta-module is labeled based on its statistical and biological properties, and a multi-label classification technique is employed to assign new disease labels to genes. We identified 3,131 gene-disease associations, validated through a literature review, Gene Ontology, and pathway analysis.

## Introduction

One of the most challenging areas of computational biology research is analyzing the complex structure of Human Disease Networks (HDN), which represent an interconnected network of intricate relationships among human diseases [1, 2]. HDNs are formed by mapping human diseases or disorders based on their genetic origins or other characteristics. In this network, two diseases are linked by an edge if they share at least one associated gene. This network typically originates from bipartite networks containing information on both genes and diseases [3]. Understanding the phenotype similarities between different diseases is important for comprehending the underlying mechanisms of these complex networks, as most human diseases are linked to groups of genes rather than a single gene [4]. Genetic heterogeneity is a major challenge in the identification of the genetic causes of complex diseases [5, 6]. Unlike

**Funding:** The author(s) received no specific funding for this work.

**Competing interests:** The author declare that they have no known competing financial interests or personal relationships that could have appeared to influence the work reported in this paper.

Mendelian diseases, which are caused by single gene mutations, complex diseases are influenced by an association of multiple genetic, environmental, and lifestyle factors. Gene modules are sets of genes that function in specific biological operations or pathways [7]. The identification of gene modules associated with diseases provides insights into the molecular mechanisms underlying them. Understanding the interactions between genes within these modules can help uncover drug targets, regulatory pathways, and biomarkers for specific complex diseases [8, 9]. Studying disease similarities is crucial for understanding underlying causes and developing treatments for complex diseases. Researchers use data mining and text mining techniques to predict and analyze disease-associated gene modules. The initial concept of the human disease network structure was proposed by Goh et al. [10]. Using the disease/disorder gene association information gathered from the Online Mendelian Inheritance in Man (OMIM: https://www.ncbi.nlm.nih.gov/omim) database, Goh et al. created a bipartite network consisting of disorder and disease genes connected by shared gene-disease relationships [11].

There are several related studies in the literature that have focused on the detection of disease-associated gene modules. In a study by Bandyopadhyay et al. [12], the authors developed a multi-objective framework to identify human protein complexes. These complexes were found to be linked to several disorder classes, including endocrine disorders, cancer, and others. Qiu et al. [13] introduced a novel network-based method that integrates gene expression and protein-protein interaction data to identify disease-associated gene modules. This approach is based on a regression model with a diffusion kernel, termed RegMOD. Wang et al. [14] proposed a novel algorithm, N2V-HC, to predict disease-associated modules based on deep learning of biological complex networks. The method involved three steps: integrating GWAS, eQTL summaries, and human interactomes; learning the node representation features in the network; and detecting modules using hierarchical clustering. Bhattacharjee et al. [15] introduced a new framework for analyzing the physical layout of disease-related PPI networks using graphlet distribution. In another study, Liu et al. [16] systematically analyzed the properties of disease proteins across 22 disorder classes. They developed two new measurement techniques, the enrichment ratio and P-value, for investigating disease proteins throughout cells/tissues and identified relationships between tissues or cells and disorder classes. Schulte-Sasse, Roman, et al. [17] proposed a machine learning method to predict 165 new cancer genes by integrating different omics datasets such as DNA methylation, gene expression and mutation with PPI networks. This EMOGI method was used to identify complex diseases and precision oncology. Hong-Dong Li et al. [18] introduced a new path for predicting genes linked to disease through the integration of various types of annotated protein sets from the molecular signature database (MSigDB) using the signal matrix as a gene feature to train the predicted disease gene model. Yang et al. [19] proposed a novel framework to identify functionally homogeneous gene modules and candidate disease-associated genes using a network-based semi-supervised learning, non-negative matrix factorization model which is called MapGene, a disease protein-prioritization algorithm. In the study, they predict the top 10 new disease genes linked with Parkinson's disease and diabetes mellitus.

In this work, we present a framework for integrating two different biological datasets aimed at predicting disease-associated modules. Unlike most previous studies that used a single dataset, the current trend in understanding cellular processes and molecular interactions involves the integration of multiple biological datasets. The integration process involves disease-associated human PPIs and Gene Ontology-based similarity, using an NMF-based clustering technique as part of our proposed framework. We detect meta-modules, which are then utilized to predict disease modules. To ensure consistency, all data sources are converted into their respective biological networks for integration purposes. Disease-associated PPI information and Gene Ontology information are converted into the disease-associated human PPI network

and the GO semantic similarity network, respectively. We create two sets of clusters or modules from these networks using the K-means clustering algorithm, which are then combined using the NMF-based clustering technique. The resulting meta-modules are classified into 22 disease or disorder classes based on their Z-scores using multi-label classification. The proposed framework for integrating biological datasets and utilizing non-negative matrix factorization (NMF), is illustrated in Fig 1. Additionally, GO and KEGG pathway enrichment analyses were conducted to evaluate the biological significance of these meta-modules. The uniqueness and motivation of the study:

- Instead of many earlier studies that used a single dataset, our study integrates three biological data sources namely gene-disease association data, human protein-protein interaction data, and Gene Ontology data.

- A non-negative matrix factorization (NMF) based clustering technique is applied here to integrate multiple biological data sources and obtain a set of meta-modules that preserve the essential characteristics of interaction patterns and functional similarity information among proteins/genes. These meta-modules are more robust and biologically relevant compared to typical single dataset analyses.

- The meta-modules are labeled based on the z-score calculated using the expected number of disease-associated genes within each obtained meta-module. We applied multi-label classification to classify and annotate the unknown genes within the meta-modules. As each meta-module takes multiple labels corresponding to one of the 22 disease classes, we first determine which genes are not linked to any of the classes. Based on the labels of genes that share the same meta-module, these genes are predicted to be associated with a disease class.

## Method

This section first discusses the workflow of the proposed method. Next, the background and formal details are described.

### Workflow

Fig 1 discusses the workflow of the proposed method. The steps are described below.

**Preprocessing of raw datasets:** See panel-A of the Fig 1. The study leverages three biological data sources: namely (i) gene-disease association data, (ii) human protein-protein interaction (PPI) data, and (iii) Gene Ontology (GO) data. We downloaded the first one from the human disease network constructed by Goh et. al. which consists of 22 diseases/disorders classes (for a more detailed view of the network see S1 Fig) and their associated genes. For the human PPI data, we downloaded the interaction information from the Human Protein Reference Database (HPRD) [20] and STRING database [21]. The gene ontology dataset was obtained from the Gene Ontology Consortium(https://geneontology.org/).

**Network construction from HDN:** See panel-A of the Fig 1. The Human Disease Network consists of 22 disease/disorder classes and 7,070 genes associated with the disease/disorders classes. For each of the disease/disorder classes, we mapped the associated genes into the HPRD and STRING databases [https://string-db.org/] [21] to construct a network. Consequently, 22 PPI networks were constructed and subsequently merged to get a large network consisting of 15,246 interactions and 1,806 unique human proteins. For the construction and merging of the network, we considered the first neighbour of each protein/gene from the HPRD and STRING databases. The network is represented as an adjacency matrix of

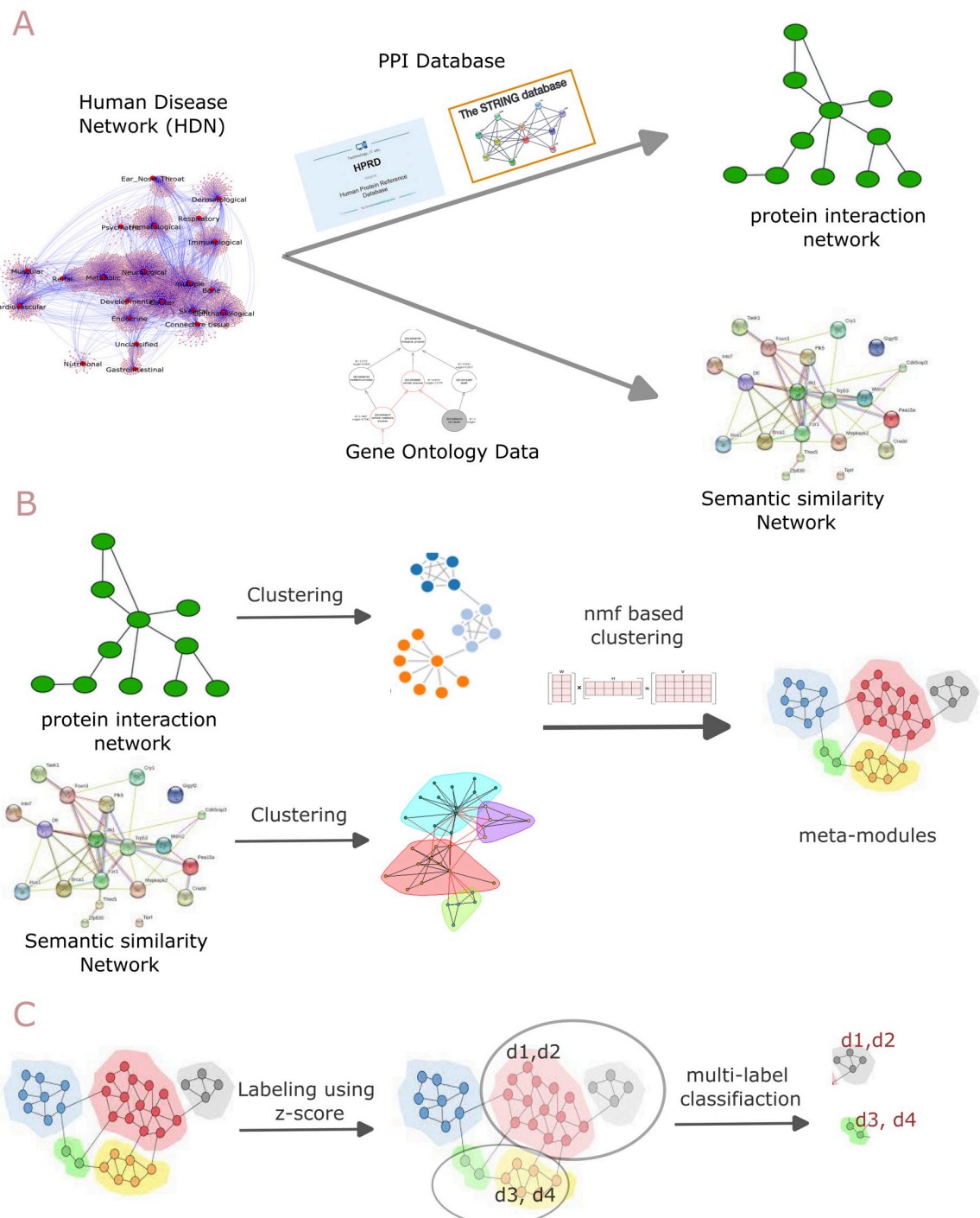

**Fig 1. Workflow describing the step-by-step process of the proposed framework.** Panel A: This panel illustrates the pre-processing of the raw datasets. The Human Disease Network consists of 22 disease/disorder classes. Their associated genes are mapped into the HPRD and STRING database to construct a PPI network and to create a semantic similarity network from Gene Ontology datasets. Panel B: the two networks (PPI Network and semantic similarity network) are partitioned using a simple k-means clustering. As a result, we got two categories of modules, which are integrated using a non-negative matrix factorization-based technique to identify meta-modules. Panel C: illustrates the labeling of meta-modules using the Z-Scores and the identification of human disease-associated gene modules through multi-label classification.

dimensions 1806 × 1806, where each entry in the matrix is represented by 0 or 1, indicating the presence or absence of a PPI connection. To construct the second network we calculate the semantic similarity score between the gene ontology terms associated with proteins/genes. Here the semantic similarity network was created using the relevance similarity measure proposed in [22]. As the score is between 0 to 1, each score was treated as the weight of an edge of the network. A high score indicates the presence of strong functional similarity between proteins/genes. For each score that was treated as the weight of an edge computation of semantic similarity, we used an R package called Csbl.go. [23]. Both of the network has the same dimensions.

**Clustering of networks and integration:** See panel-B of the Fig 1. The two networks (PPI network and semantic similarity network) are partitioned using a simple k-means clustering, with the number of clusters (k) determined by Silhouette Analysis (For more information, see S2 Fig). From the PPI network, four clusters are identified and from the semantic similarity network, 6 clusters are identified by the clustering analysis. Thus, two categories of modules are obtained, one derived from interaction information and the other from ontology information. The two categories of modules are then integrated by using a matrix factorization-based framework. Here, we use a non-negative matrix factorization technique [24] to fuse these two information obtaining a set of meta-modules (see section [Formal details and background] for a detailed description of nmf-based clustering). Using silhouette analysis, seven meta-modules are identified (details information, see S3 Fig).

**Labeling meta-modules:** The obtained meta-modules are labeled based on the Z–Score calculated using the expected number of disease-associated genes within each obtained meta-module. We calculate $Z - Score$ of each meta module $m_i$ and for each disease class $d_j$ as: $Z\_Score_{m_i,d_j} = (G_{m_i,d_j} - E_{m_i})/SQRT(E_{m_i})$, where $|G_{m_i,d_j}|$ is the total number of genes of module $m_i$ associated with disease $d_j$, $|E_{m_i}|$ represents expected number of genes of module $m_i$ associated with the disease $d_j$. $|G_{m_i}|$ is obtained by computing the intersection between the genes of modules $m_i$ with the genes associated with disease $d_j$. For $|E_{m_i}|$, we take the expected number of genes associated with a specific disease $d_j$ that should be present within a module $m_i$. This is written as $|E_{m_i}| = \frac{|G_{m_i}| \times |G_{d_j}|}{N}$, where $|G_{m_i}|$ is number of genes within a meta-module $m_i$, whereas $|G_{d_j}|$ is the number of genes associated with disease $d_j$ and $N$ represents total genes associated with all diseases.

**Identification of novel associations:** See the panel-C of the Fig 1. We utilized multi-label classification to classify and annotate the unidentified genes within the meta-modules. Since each meta-module receives multiple labels corresponding to one of the 22 disease classes, we first identify genes that are not associated with any of these classes. These genes are then predicted to be associated with the disease classes corresponding to the labels of genes sharing the same meta-module.

## Formal details and background

Here we have discussed the formal details and background of the developed methods.

### Gene ontology-based semantic similarity

Gene Ontology(GO)-based semantic similarity (SS) is a pivotal concept in bioinformatics, allowing the comparison of GO terms or entities annotated with these terms by leveraging the ontology's structure, properties, and annotation corpora. Over the past decade, the diversity and number of SS measures based on GO have significantly increased. These measures are applied in various domains, including functional coherence evaluation, protein interaction

prediction, and disease gene prioritization [25–27]. A semantic similarity measure is characterized as a function which, when provided with two ontology terms or collections of terms annotating two entities, produces a numeric outcome indicating the proximity in meaning between them. This measure can be calculated for a pair of GO terms, referred to as term similarity, or for two gene products, each annotated with a set of GO terms, referred to as gene product similarity.

In the context of Gene Ontology (GO), two primary approaches are used to measure Semantic Similarity (SS): internal methods based on ontology structure and external methods based on external corpora. The simplest structural methods calculate the distance between two nodes as the number of edges in the path between them. This edge-counting approach is intuitive but assumes that all semantic links have the same weight, thereby defining SS as the inverse score of the semantic distance.

External methods frequently employ information-theoretic principles, which are less affected by variations in link density within the ontology graph. Measures based on information content (IC) stem from the concept that the similarity between two concepts can be gauged by how much information they share.

When dealing with gene products annotated with multiple GO terms, measuring gene product semantic similarity (SS) involves comparing sets of terms. Two primary strategies are commonly employed: pairwise and group-wise. Pairwise methods aggregate individual similarities between all terms annotating two gene products to derive an overall measure of functional similarity. Group-wise methods, on the other hand, directly compute gene product similarity using one of three approaches: set-based, graph-based, or vector-based. Set-based approaches consider only direct annotations and may overlook shared ancestry among GO terms. Graph-based approaches represent gene products as sub-graphs of the GO, incorporating both direct and inherited annotations for a more comprehensive model. Vector-based approaches represent gene products in vector space and determine functional similarity using vector similarity measures. Group-wise methods can also incorporate the Information Content (IC) of terms to weigh set similarity effectively.

This comprehensive understanding and selection of SS measures are crucial for biomedical researchers to choose the most appropriate approaches for their specific applications, especially given the increased complexity of GO and the need for efficient computation in the future generation of SS measures.

## NMF based clustering

**NMF.**   Non-negative Matrix Factorization (NMF) is a dimensionality reduction technique widely used in various fields such as signal processing, image analysis, and topic modelling [28, 29]. It involves decomposing a given non-negative matrix into the product of two lower-dimensional matrices, both of which contain only non-negative values. NMF is particularly useful when dealing with data where the non-negativity constraint is meaningful, such as in image or text data.

The general idea of NMF can be expressed using the following equation for a given matrix $Z$ of size $A \times B$: $Z \approx PQ^T$. Here, $Z$ is the original matrix, $P$ is an $A \times K$ matrix, $Q$ is a $B \times K$ matrix, and $K$ is the reduced dimensionality. The aim is to find the matrices $P$ and $Q$ such that their product approximates $Z$ almost approximately, and $K$ is typically chosen to be much smaller than both $A$ and $B$.

The matrices $P$ and $Q$ are subject to the constraint that all their elements must be non-negative ($P, Q \geq 0$)). This non-negativity constraint is crucial in applications where negative values do not have a meaningful interpretation.

The approximation error, or the difference between the original matrix $Z$ and its approximation $PQ^T$, can be minimized using various optimization techniques. One common approach is to use iterative algorithms, such as multiplicative updates or gradient descent, to adjust the values in $P$ and $Q$ until the approximation is satisfactory.

The NMF process can be mathematically formulated as an optimization problem. Given a cost function $C$, the objective is to minimize $C$ with respect to $P$ and $Q$, subject to the non-negativity constraint:

$$\min_{P,Q} C(P, Q) \quad \text{subject to} \quad P, Q \geq 0$$

The cost function $C$ typically measures the difference between $Z$ and $PQ^T$, such as the Frobenius norm or Kullback-Leibler divergence. The choice of $K$ determines the reduced dimensionality of the factorization. The optimization process involves minimizing a cost function that quantifies the approximation error.

**Integration of clustering using NMF.** The integration process aimed to select a set of meta-modules comprising genes/proteins co-occurring in two distinct categories of modules, thereby preserving the characteristics of two different data resources within these meta-modules. Fig 2 illustrates the integration process of modules from different categories. Generally, when dealing with multiple networks offering diverse perspectives, it becomes possible to

**Fig 2. Illustrates the integration process of meta-modules from different biological sources of information.** In the first step, the protein-protein interaction (PPI) data and GO-based semantic similarity data are clustered using the K-means clustering technique to obtain two categories of modules or clusters. Subsequently, these two types of modules are integrated through NMF-based techniques to generate a set of meta-modules.

extract modules that retain the unique characteristics of each view. The process begins with the clustering of networks corresponding to each view. The resulting clusters are then combined to generate a co-occurrence matrix that preserves the occurrence of nodes within modules of different categories. In Fig 2, genes $g_i$ and $g_j$ share the same module three times, resulting in a co-occurrence score of three in the matrix. The co-occurrence matrix ($Z$) is subsequently decomposed using NMF-based factorization techniques, denoted as $Z \approx PQ^T$. Here, $P \in \mathbb{R}^{n \times k}$ represents the projection of the original data onto a new set of basis vectors, also referred to as meta-cluster centroids, where $k$ is the number of meta-clusters. Here, the silhouette Analysis determines the value of k, and seven meta-modules are identified by the clustering analysis. These meta-clusters can be additively combined using the columns of matrix $Q \in \mathbb{R}^{k \times m}$. To measure the reconstruction error between the original data and the factors $P$ and $Q$, the Frobenius norm is utilized.

In this study, two different views of the gene nodes were considered: interaction information and gene ontology-based semantic similarity information (see Fig 1 for details).

## Multi-label classification

Multi-label classification is a machine learning paradigm where instances can be simultaneously associated with multiple labels, in contrast to the traditional single-label classification [30]. In the latter, each instance is exclusively linked to a single class label whether multi-class or binary. The multi-label paradigm has garnered increasing attention and finds applicability across diverse domains, including text, audio data, still images, video, bioinformatics, and others.

The prevailing method in multi-label classification involves training independent classifiers for each label, commonly referred to as the binary relevance (BR) transformation. Essentially, this approach transforms a multi-label problem into multiple binary problems, with each label treated independently. Existing multi-label literature acknowledges the limitation of this method due to its failure to explicitly model dependencies between labels. Consequently, various algorithms have been proposed to address this limitation by incorporating and modelling label dependencies explicitly.

The label space is denoted as $\mathcal{L}$, encompassing all possible labels. For a given instance, the output is a subset $Y \subseteq \mathcal{L}$ signifying the presence of specific labels. The decision function $f_l(X)$ assigns a binary value to each label $l \in \mathcal{L}$ based on the input features $X$, with 1 indicating presence and 0 indicating absence. The relationship between instances and labels is modelled by a binary indicator matrix $Y$, where $Y_{ij} = 1$ if label $j$ is present for instance $i$, and $Y_{ij} = 0$ otherwise. A common strategy, known as Binary Relevance, involves training separate binary classifiers for each label independently. The final prediction $Y_i$ for an instance is obtained as the union of individual label predictions ($Y_i = \bigcup_{l \in \mathcal{L}} f_l(X_i)$).

Hamming Loss is a commonly used evaluation metric in multi-label classification tasks. It measures the fraction of labels that are incorrectly predicted for a given set of instances. In the context of multi-label classification, where an instance can belong to multiple classes simultaneously, Hamming Loss is particularly relevant as it considers each label independently.

The Hamming Loss ($H$) is calculated as the average fraction of incorrect labels over all instances. If $Y_{ij}$ represents the true binary label for instance $i$ and label $j$, and $\hat{Y}_{ij}$ represents the predicted binary label, then the Hamming Loss for a set of instances is given by:

$$H = \frac{1}{N} \sum_{i=1}^{N} \frac{1}{|\mathcal{L}|} \sum_{j=1}^{|\mathcal{L}|} (\hat{Y}_{ij} \neq Y_{ij})$$

Here, $N$ is the total number of instances, $|\mathcal{L}|$ is the total number of labels, and $\neq$ denotes the inequality operator.

The Hamming Loss ranges from 0 to 1, where lower values indicate better performance. A Hamming Loss of 0 means perfect predictions, while a value of 1 indicates that all labels are incorrectly predicted.

Multi-label classification finds diverse applications in domains such as image classification (assigning multiple object labels to an image), text categorization (assigning multiple topics to a document), and bioinformatics (predicting multiple gene functions). This approach provides a flexible and nuanced understanding of complex data relationships by allowing instances to be associated with multiple labels simultaneously.

## Results and discussions

In this section, experimental results are described.

### Datasets used

The study leverages three biological data sources, namely (i) gene-disease association data, (ii) human protein-protein interaction data and (iii) Gene Ontology data. We have compiled a comprehensive list of human PPIs from two datasets: (1) CCSB human Interactome database consisting of 7,000 proteins and 13944 high-quality binary interactions [31–33], (2) The Human Protein Reference Database [34] consisting of 8920 proteins and 53184 PPIs. We collected Gene Ontology information of those selected proteins from the GO Consortium (http://www.geneontology.org/) [35]. However, the protein IDs in the PPI dataset are different from those in the GO dataset, which requires mapping between the two. We used the David Bioinformatics resource (https://david.ncifcrf.gov/home.jsp) [36] tool for protein IDs conversion. We then took the entire gene IDs in the HPRD dataset that match with the UniProt IDs in the GO database for a specific gene symbol.

HDN was created by Goh et al. with 22 disease/disorder classes and their associated genes mapped into the HPRD and STRING databases to create 22 PPI networks, which were then merged to get a network containing 15,246 interactions and 1806 unique human genes.

### Labeling of meta-modules with disease class

The GO database was used to collect gene ontology information on these selected genes, and a semantic similarity network was created based on them. An adjacency and semantic similarity matrix were constructed and partitioned into four and six clusters using the K-means clustering technique. Thereafter, two modules or clusters are integrated to form meta-modules by using the NMF clustering technique.

From the PPI network four clusters and from the semantic similarity network 6 clusters are identified by the clustering analysis. Thus two categories of modules are obtained, one from interaction information and the other from ontology information. The two categories of modules are then integrated by using a matrix factorization-based framework.

The obtained meta-modules are labeled based on the z-score calculated using the expected number of disease-associated genes within each obtained meta-modules. Thus each meta module gets a z-score based on the expected number of disease-associated genes within it. Now a threshold is selected to label a meta module to one of the disease classes. The figure and table (see S4 Fig and S1 Table in supplementary for details) show the labeling of the meta-modules in the different disease classes. For example, meta-module-2 is predicted to be included in seven of the disease classes (Neurological, Ophthalmological, Skeletal, Cardiovascular, Cancer, and Endocrine). The genes within the meta-module are also labeled with the same disease

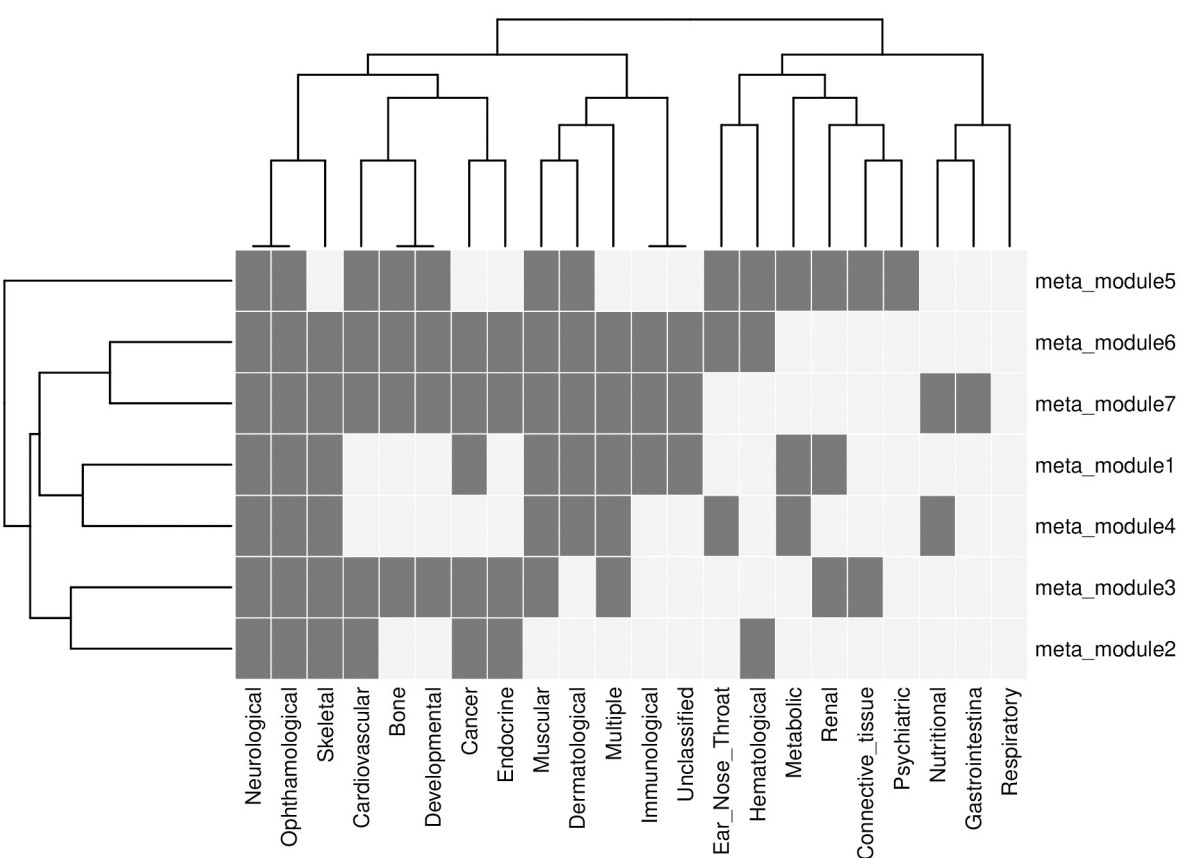

**Fig 3. Heatmap representing the membership of meta-modules within the disease classes.** For example, meta-module-2 is associated with seven of the disease classes (Neurological, Ophthalmological, Skeletal, Cardiovascular, Cancer, and Endocrine).

class. Fig 3, shows a heatmap representing the membership of the meta-modules within each of the disease classes.

## Prediction of gene-disease association

The labeled network was then trained using multi-label classification. The input was the concatenation of the adjacency and semantic similarity matrix of all the nodes. We have utilized the 'makeMultilabelClassifierChainsWrapper' function of mlr R [37] package for performing the classification. The Table 1 shows the multi-label Hamming loss of the classification results. The trained model is utilized to predict the class label of unknown genes. We have obtained 3131 genes associated with multiple disease classes. Fig 4 depicts the predicted gene-disease

**Table 1. The results of the classifiers for multi-label, multi-class dataset.**

| Sl No. | Classifier | Hamming-loss |
|---|---|---|
| 1 | Decision tree | 0.045998446 |
| 2 | SVM | 0.046245685 |
| 3 | Random Forest | 0.045298344 |
| 4 | KNN | 0.046930847 |

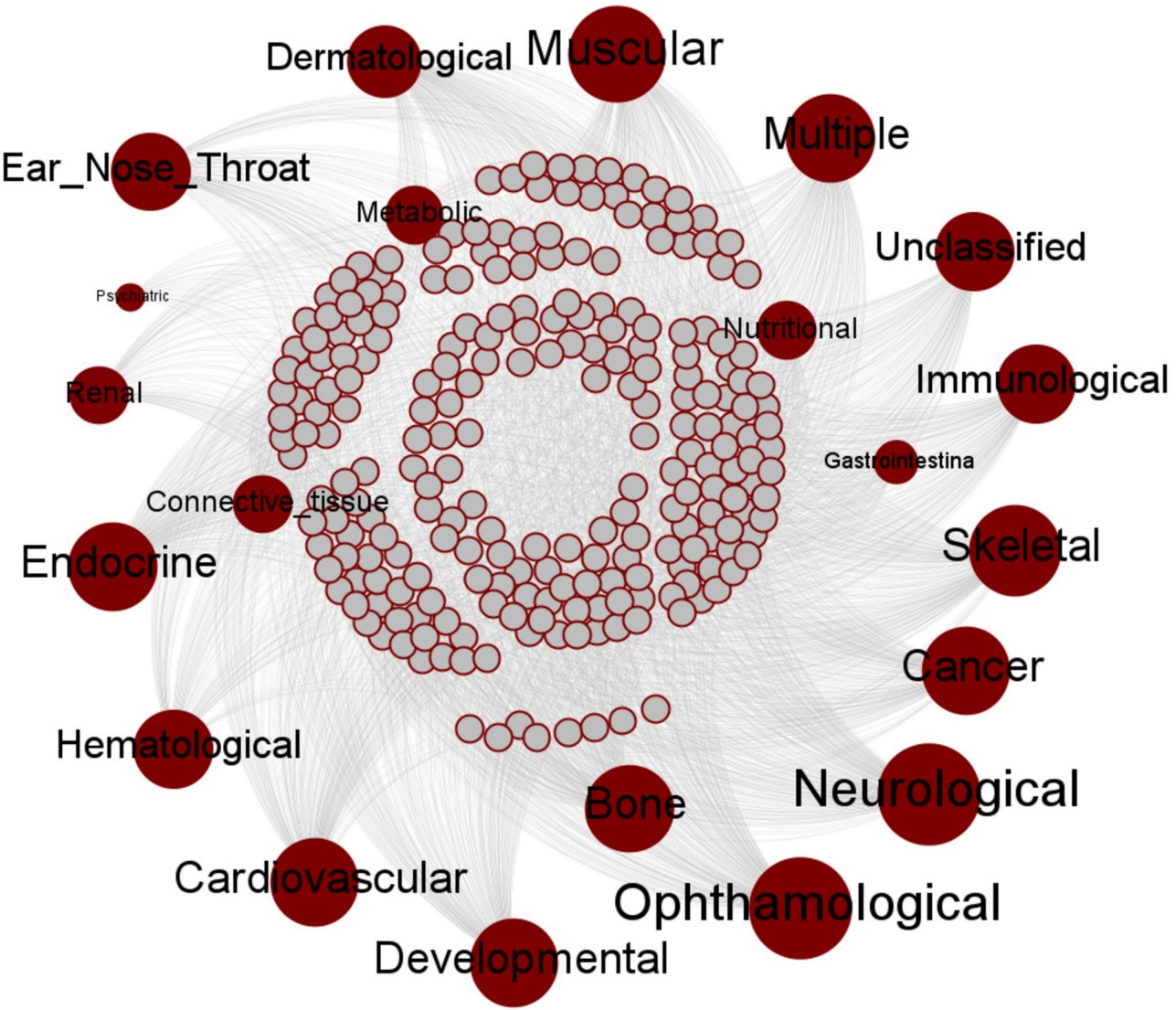

**Fig 4. Predicted disease-gene interaction network consisting of genes and their associated disease/disorder classes.**

interaction network, illustrating how genes are associated with various disease classes. This network helps visualize potential novel interactions and supports further validation and exploration of these relationships. Table 2 lists some of the predicted gene-disease associations along with their supporting PubMed IDs, demonstrating the validity of these associations based on existing literature. The last column of Table 2 displays the PubMed IDs of the literature supporting the predicted associations. For instance, the association of the gene "MTHFD1" with cancer is supported by PubMed ID 35798877, indicating consistency with recent studies in cancer biology.

## Meta-modules are enriched with GO terms and pathways

To know the efficacy of the prediction of the genes within the meta-modules which are predicted to be associated with different disease classes we have performed a gene ontology-based

**Table 2. Predicted interactions between disease/disorder classes and human proteins.** PUBMED ID is shown as validation of the identified predicted interactions.

| Sl No. | Disease Classes | Human Protein | PUBMED ID |
|---|---|---|---|
| 1 | Cancer | MTHFD1 | 35798877 |
| 2 | Cancer | GPR143 | 36800996 |
| 3 | Cancer | HSPA14 | 34765552 |
| 4 | Cancer | S100A4 | 29069865 |
| 5 | Cancer | ABCG2 | 28005280 |
| 6 | Cancer | ALDH9A1 | 22206977 |
| 7 | Cancer | COTL1 | 35873386 |
| 8 | Bone | ELANE | 33513358 |
| 9 | Bone | MECOM | 37407873 |
| 10 | Developmental | HNRNPU | 35274911 |
| 11 | Developmental | MED13L | 36798993 |
| 12 | Renal | PCBD1 | 30175537 |
| 13 | Renal | HADH | 34391866 |
| 14 | Renal | HNF1B | 25536396 |
| 15 | Metabolic | CYP2D6 | 33143137 |
| 16 | Metaboli | MOGS | 35575968 |
| 17 | Metabolic | ADIPOQ | 36404807 |
| 18 | Metabolic | FOXO1 | 37941908 |
| 19 | Ophthamological | VSX2 | 37905144 |
| 20 | Ophthamological | GPR98 | 23441107 |
| 21 | Ophthamological | PRPF6 | 38074011 |
| 22 | Muscular | FKTN | 35810429 |
| 23 | Cardiovascular | NR4A3 | 34768801 |
| 24 | Cardiovascular | NKX2–5 | 38124890 |
| 25 | Neurological | IL1RAPL1 | 23785489 |
| 26 | Neurological | ATN1 | 36007104 |
| 27 | Neurological | VEGFA | 34713920 |
| 28 | Immunological | HLA-DRB1 | 29037901 |
| 29 | Skeletal | AGGF1 | 36696895 |
| 30 | Multiple | CHIT1 | 31997416 |

analysis of all the meta-modules. The most significant and practical techniques for understanding the biological significance of the observed expression change are gene ontology and pathway-based analysis. This analysis reveals that meta-modules are significantly enriched with terms related to various biological processes and pathways, further supporting their relevance in disease contexts. Table 3 presents the enriched GO terms and pathways associated with each identified meta-module, providing insights into the biological significance of these modules. For example, meta-module 1 is enriched with the term "visual perception" (GO:0007601) with a $p-value$ of $1.35E-59$, suggesting its potential role in ophthalmological diseases.

## A short discussion of key findings and their theoretical implications

The study's findings indicate that the integration of diverse biological data sources using NMF-based clustering can effectively identify novel gene-disease associations. The multi-label classification approach demonstrated strong predictive capabilities, with a high level of agreement between predicted associations and existing literature. These results support the theory

**Table 3. Table shows significant Gene Ontology (GO), $P-values$ and KEGG Pathway associated with the identified meta-modules.**

| Sl No. | No. of Genes | Gene ID | GO Term | KEGG Pathway |
|---|---|---|---|---|
| 1 | 142 | GO:0007601 | visual perception (1.35E-59) | Metabolic pathways (1.04E-34) |
| 2 | 70 | GO:0007601 | visual perception (3.52E-60) | Phototransduction (8.05E-12) |
| 3 | 254 | GO:0009410 | response to xenobiotic stimulus (3.33E-33) | Lysosome (2.45E-13) |
| 4 | 217 | GO:0008284 | positive regulation of cell proliferation (8.05E-47) | Pathways in cancer (2.17E-25) |
| 5 | 219 | GO:0045944 | positive regulation of transcription from RNA polymerase II promoter (6.31E-227) | Transcriptional misregulation in cancer (2.79E-16) |
| 6 | 541 | GO:0006936 | muscle contraction (2.13E-11) | Hypertrophic cardiomyopathy (4.93E-08) |
| 7 | 363 | GO:0016558 | protein import into peroxisome matrix (3.06E-08) | Protein digestion and absorption (3.14E-07) |

that multiple biological datasets when integrated, can provide a more comprehensive understanding of disease mechanisms and gene functions.

The identification of meta-modules enriched with significant GO terms and pathways suggests that these modules represent functionally coherent groups of genes that may play critical roles in the development and progression of various diseases. This challenges the traditional view that single datasets are sufficient for understanding complex disease biology and supports the growing consensus in the field that integrative approaches are essential for uncovering hidden biological insights.

## Conclusions

In this paper, we introduce a novel framework for predicting disease-associated modules by integrating multiple biological data sources. The study leverages disease-associated protein-protein interactions (PPIs) and Gene Ontology-based similarity as information sources, which are integrated using non-negative matrix factorization (NMF) based clustering. The data sources are converted into respective biological networks and integrated through the NMF-based clustering method. The resulting clusters, termed as meta-modules, are further classified into 22 diseases/disorders using a multi-label classification based on their Z-Scores.

To address the multi-label nature of the data, we employ the binary-relevance method, transforming the problem into multiple binary classification tasks. We evaluate various classifiers and identify the combination of binary relevance and the K-nearest neighbor (KNN) classifier as the most effective in predicting disease-associated modules. This study demonstrates the development of innovative computational techniques for analyzing large biological networks.

By integrating diverse biological datasets and employing advanced clustering and classification methods, the proposed framework provides valuable insights into disease-gene associations and lays a strong foundation for future research in precision medicine and disease diagnosis. This approach holds significant potential for enhancing our understanding of complex biological interactions and identifying key factors contributing to various diseases and disorders.

However, it is important to acknowledge the limitations of this study, including the quality and completeness of available data and the need for further experimental validation of the predicted associations. Future research should focus on expanding the data integration to include additional omics datasets, improving the methods for capturing dynamic interactions over time, and validating the findings through experimental studies. This will help refine the framework, extend its applications in personalized medicine, improve disease management

strategies, and pave the way for novel therapeutic interventions, ultimately advancing the field of network and precision medicine.

## Supporting information

**S1 Fig. The Human Disease Network.** The figure illustrates a graphical representation of the Human Disease Network (HDN with disease or disorder classes represented by one set of nodes and interacted genes represented by another set of nodes.
(TIF)

**S2 Fig. Finding the optimal number of clusters using the average silhouette method.** The two networks (PPI Network and semantic similarity network) are partitioned using a simple k-means clustering with k determined by the Silhouette Analysis.
(TIF)

**S3 Fig. The silhouette analysis determines the value of k, and seven meta-modules are identified by the clustering analysis.**
(TIF)

**S4 Fig. Z_score based labeling of meta–modules across different disease classes.**
(TIF)

**S1 Table. The table shows Z_score value of each meta–modules in the different disease classes.**
(TIF)

## Author Contributions

**Conceptualization:** Sumanta Ray.

**Data curation:** Syed Alberuni.

**Formal analysis:** Syed Alberuni, Sumanta Ray.

**Investigation:** Sumanta Ray.

**Methodology:** Syed Alberuni, Sumanta Ray.

**Project administration:** Sumanta Ray.

**Resources:** Syed Alberuni, Sumanta Ray.

**Software:** Syed Alberuni, Sumanta Ray.

**Supervision:** Sumanta Ray.

**Validation:** Syed Alberuni, Sumanta Ray.

**Visualization:** Syed Alberuni, Sumanta Ray.

**Writing – original draft:** Syed Alberuni.

**Writing – review & editing:** Sumanta Ray.

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
