## [Decision Letter · Decision Letter 0]

19 Jul 2024

PONE-D-24-22050Integration of Biological Data via NMF for Identification of Human Disease-Associated Gene Modules through Multi-label Classification.PLOS ONE

Dear Dr. ALBERUNI,

Thank you for submitting your manuscript to PLOS ONE. After careful consideration, we feel that it has merit but does not fully meet PLOS ONE’s publication criteria as it currently stands. Therefore, we invite you to submit a revised version of the manuscript that addresses the points raised during the review process.

We look forward to receiving your revised manuscript.

Kind regards,

Zhu-Hong You

Academic Editor

PLOS ONE

Additional Editor Comments:

19-JUL-2024

Manuscript ID: PONE-D-24-22050

Dear Dr. SYED ALBERUNI :

Your manuscript "Integration of Biological Data via NMF for Identification of Human Disease-Associated Gene Modules through Multi-label Classification." which you submitted to the PLOS ONE, has been reviewed. The comments of the reviewer(s) are included at the bottom of this letter. Please respond to the reviewer(s)' comments and revise your manuscript.

Reviewer 1: The article titled "Integration of Biological Data via NMF for Identification of Human Disease-Associated Gene Modules through Multi-label Classification" aims to integrate biological data using Non-negative Matrix Factorization (NMF) to identify human disease-associated gene modules. The content is rich but can be further improved in certain aspects. Here are some suggestions:

1.Clarity and Conciseness of the Abstract: The abstract is somewhat lengthy. It is recommended to simplify the language and highlight the core findings and methods of the study. For example: Instead of "Extensive evidence recognizes that proteins associated with several diseases frequently interact with each other...", it can be summarized as "Proteins associated with multiple diseases often interact, forming disease modules critical for understanding disease mechanisms. This study integrates biological data using NMF to identify such modules."

Background Introduction in the Introduction Section:

2.The uniqueness and innovation of this study should be highlighted to capture the reader's interest.

3.Detailed Methodology Section: The methodology section provides detailed descriptions of the NMF technique and multi-label classification, but more information or references on technical details such as parameter selection and algorithm implementation can be provided. The explanations of Figures 1 and 2 can be more detailed to ensure readers can independently understand the illustrations.

4.Data Presentation in the Results Section: The results section includes numerous tables and illustrations, such as Table 1, Table 2, and Figure 3. It is recommended to add brief descriptions after each table and figure to help readers better understand the significance of the data.

Additionally, more discussion on key findings should be included in the results section, explaining how these findings support or challenge existing theories.

5.In-depth Analysis in the Discussion and Conclusion Sections: The discussion section can be more in-depth. Besides summarizing the findings, it should also discuss the limitations of the study and future research directions. The conclusion section can be more concise, highlighting the contributions of the study and its potential in practical applications.

6.Consistency in Language and Format:Check for grammatical and spelling errors throughout the text to ensure language accuracy and professionalism.Ensure the format of the references is consistent, and that all cited literatures are fully detailed in the reference list.

Reviewer 2: The manuscript introduces a novel approach utilizing non-negative matrix factorization (NMF) to integrate protein-protein interactions (PPIs) and Gene Ontology data for discovering disease modules. The study clusters networks derived from these sources into modules and integrates them through an NMF-based technique to identify meta-modules, facilitating gene-disease association predictions. The manuscript could be accepted after solving the following suggestion.

1. The review of related work is not sufficiently thorough and not sufficiently specific. There are too few examples in the introduction. It is suggested to add more state-of-the-art research work from predecessors.

2. The authors should ensure that the English language in this manuscript meets our high standards, is grammatically correct, and conforms to professional standards of courtesy and expression.

3. The computational details are not mentioned. Which software, program languages, libraries, etc., were used to build this approach? Is it possible to publish code to use this architecture and reproduce the results?

4. The motivation part can be elaborated in the abstract, in the introduction, and the Discussion, if possible, which will enable the authors to improve the academic merit of their work.

5. The authors only mentioned the results in the Discussion section, while there is no in-depth discussion.

6. The figures and Tables quality needs to be improved.

7. Some technical aspects and essential insights of the proposed method are not described in detail.

Reviewers' comments:

Reviewer's Responses to Questions

**Comments to the Author**

1. Is the manuscript technically sound, and do the data support the conclusions?

Reviewer #1: Yes

Reviewer #2: Yes

2. Has the statistical analysis been performed appropriately and rigorously? 

Reviewer #1: Yes

Reviewer #2: Yes

3. Have the authors made all data underlying the findings in their manuscript fully available?

Reviewer #1: Yes

Reviewer #2: No

4. Is the manuscript presented in an intelligible fashion and written in standard English?

Reviewer #1: Yes

Reviewer #2: Yes

5. Review Comments to the Author

Reviewer #1: The article titled "Integration of Biological Data via NMF for Identification of Human Disease-Associated Gene Modules through Multi-label Classification" aims to integrate biological data using Non-negative Matrix Factorization (NMF) to identify human disease-associated gene modules. The content is rich but can be further improved in certain aspects. Here are some suggestions:

1.Clarity and Conciseness of the Abstract: The abstract is somewhat lengthy. It is recommended to simplify the language and highlight the core findings and methods of the study. For example: Instead of "Extensive evidence recognizes that proteins associated with several diseases frequently interact with each other...", it can be summarized as "Proteins associated with multiple diseases often interact, forming disease modules critical for understanding disease mechanisms. This study integrates biological data using NMF to identify such modules."

Background Introduction in the Introduction Section:

2.The uniqueness and innovation of this study should be highlighted to capture the reader's interest.

3.Detailed Methodology Section: The methodology section provides detailed descriptions of the NMF technique and multi-label classification, but more information or references on technical details such as parameter selection and algorithm implementation can be provided. The explanations of Figures 1 and 2 can be more detailed to ensure readers can independently understand the illustrations.

4.Data Presentation in the Results Section: The results section includes numerous tables and illustrations, such as Table 1, Table 2, and Figure 3. It is recommended to add brief descriptions after each table and figure to help readers better understand the significance of the data.

Additionally, more discussion on key findings should be included in the results section, explaining how these findings support or challenge existing theories.

5.In-depth Analysis in the Discussion and Conclusion Sections: The discussion section can be more in-depth. Besides summarizing the findings, it should also discuss the limitations of the study and future research directions. The conclusion section can be more concise, highlighting the contributions of the study and its potential in practical applications.

6.Consistency in Language and Format:Check for grammatical and spelling errors throughout the text to ensure language accuracy and professionalism.Ensure the format of the references is consistent, and that all cited literatures are fully detailed in the reference list.

Reviewer #2: The manuscript introduces a novel approach utilizing non-negative matrix factorization (NMF) to integrate protein-protein interactions (PPIs) and Gene Ontology data for discovering disease modules. The study clusters networks derived from these sources into modules and integrates them through an NMF-based technique to identify meta-modules, facilitating gene-disease association predictions. The manuscript could be accepted after solving the following suggestion.

1. The review of related work is not sufficiently thorough and not sufficiently specific. There are too few examples in the introduction. It is suggested to add more state-of-the-art research work from predecessors.

2. The authors should ensure that the English language in this manuscript meets our high standards, is grammatically correct, and conforms to professional standards of courtesy and expression.

3. The computational details are not mentioned. Which software, program languages, libraries, etc., were used to build this approach? Is it possible to publish code to use this architecture and reproduce the results?

4. The motivation part can be elaborated in the abstract, in the introduction, and the Discussion, if possible, which will enable the authors to improve the academic merit of their work.

5. The authors only mentioned the results in the Discussion section, while there is no in-depth discussion.

6. The figures and Tables quality needs to be improved.

7. Some technical aspects and essential insights of the proposed method are not described in detail.

6. PLOS authors have the option to publish the peer review history of their article (what does this mean?). If published, this will include your full peer review and any attached files.

Reviewer #1: No

Reviewer #2: No

---

## [Author Response · Author response to Decision Letter 0]

10 Sep 2024

Reviewer#: 1

The article titled "Integration of Biological Data via NMF for Identification of Human Disease-Associated Gene Modules through Multi-label Classification" aims to integrate biological data using Non-negative Matrix Factorization (NMF) to identify human disease-associated gene modules. The content is rich but can be further improved in certain aspects. Here are some suggestions:

1. Clarity and Conciseness of the Abstract: The abstract is somewhat lengthy. It is recommended to simplify the language and highlight the core findings and methods of the study. For example: Instead of "Extensive evidence recognizes that proteins associated with several diseases frequently interact with each other...", it can be summarized as "Proteins associated with multiple diseases often interact, forming disease modules critical for understanding disease mechanisms. This study integrates biological data using NMF to identify such modules."

Answer: We have now revised the abstract to emphasize the key findings and methodology of our study. Please see the updated abstract in the revised version of the manuscript.

Background Introduction in the Introduction Section:

2. The uniqueness and innovation of this study should be highlighted to capture the reader's interest.

Answer: As suggested by the reviewer we have now included the uniqueness and motivation of this study in the last paragraph of the section Introduction. Please see page no. 03 of the revised version of the manuscript.

i) Instead of many earlier studies that used a single dataset, our study integrates three biological data sources namely gene-disease association data, human protein-protein interaction data and Gene Ontology data.

ii) A non-negative matrix factorization(NMF) based clustering technique is applied here to integrate multiple biological data sources and obtain a set of meta-modules

that preserve the essential characteristics of interaction patterns and functional similarity information among the proteins/genes. These meta-modules are more robust and biologically relevant compared to typical single dataset analyses.

iii) The meta-modules are labelled based on the z-score calculated using the expected number of disease-associated genes within each obtained meta-module. We applied multi-label classification to classify and annotate the unknown genes within meta-modules. As each meta-module takes multiple labels corresponding to one of the 22 disease classes, we first determine which genes are not linked to any of the classes. Based on the labels of genes that share the same meta-module, these genes are then predicted to be associated with the disease class.

3. Detailed Methodology Section: The methodology section provides detailed descriptions of the NMF technique and multi-label classification, but more information or references on technical details such as parameter selection and algorithm implementation can be provided. The explanations of Figures 1 and 2 can be more detailed to ensure readers can independently understand the illustrations. 

Answer: We have expanded the methodology section to include additional information on parameter selection and algorithm implementation for both the NMF technique and multi-label classification. Specifically, we now provide more detailed descriptions of how parameters were chosen for each algorithm, such as the use of silhouette analysis for determining the optimal number of clusters (k). This analysis was conducted for both the K-means clustering and NMF algorithms, and the value of k was selected based on the highest silhouette score. Further details, including visualizations of the silhouette analysis, are provided in the supplementary figures to aid in the understanding of this process. For the multi-label classification, we used the Utiml package in R, which provides various tools for multi-label learning. Specifically, we utilized the 'makeMultilabelClassifierChainsWrapper' function from the mlr R package to perform classification. We also improved the descriptions of Figures 1 and 2 to ensure that readers can follow the illustrations independently.

4. Data Presentation in the Results Section: The results section includes numerous tables and illustrations, such as Table 1, Table 2, and Figure 3. It is recommended to add brief descriptions after each table and figure to help readers better understand the significance of the data.

Additionally, more discussion on key findings should be included in the results section, explaining how these findings support or challenge existing theories.

Answer: We have added brief descriptions following each table and figure in the results section to enhance clarity and help readers understand the significance of the presented data.

Additionally, we have expanded the discussion of key findings, explaining how they support or challenge existing theories and contribute to the understanding of disease-associated gene modules. Moreover, we have included a subsection ‘A short discussion of key findings and their theoretical implication’ to provide a summary of key findings and their implications. Please see the last paragraph of each subsection in the Results section for the expanded discussion, and refer to the newly added subsection "A Short Discussion of Key Findings and Their Theoretical Implications" in the Results section for a comprehensive summary.

5. In-depth Analysis in the Discussion and Conclusion Sections: The discussion section can be more in-depth. Besides summarizing the findings, it should also discuss the limitations of the study and future research directions. The conclusion section can be more concise, highlighting the contributions of the study and its potential in practical applications.

Answer: We have expanded the discussion section to provide a more in-depth analysis, including a summary of the key findings, explaining how they support or challenge existing theories and contribute to the understanding of disease-associated gene modules.

. Additionally, we have revised the conclusion section to be more concise, clearly highlighting the study's contributions and limitations and its potential applications in understanding disease mechanisms and advancing precision medicine. Please see the last paragraph of each subsection in the Results section for the expanded discussion, and refer to the newly added subsection "A Short Discussion of Key

Findings and Their Theoretical Implications" in the Results section for a comprehensive summary. Additionally please see the last paragraph of the Conclusions section of the revised version of the manuscript.

6. Consistency in Language and Format: Check for grammatical and spelling errors throughout the text to ensure language accuracy and professionalism. Ensure the format of the references is consistent, and that all cited literatures are fully detailed in the reference list.

Answer: We have thoroughly reviewed the manuscript for grammatical and spelling errors to enhance language accuracy and professionalism. Additionally, we have ensured consistency in the reference format and verified that all cited literature is fully detailed in the reference list.

Reviewer#: 2

The manuscript introduces a novel approach utilizing non-negative matrix factorization (NMF) to integrate protein-protein interactions (PPIs) and Gene Ontology data for discovering disease modules. The study clusters networks derived from these sources into modules and integrates them through an NMF-based technique to identify meta-modules, facilitating gene-disease association predictions. The manuscript could be accepted after solving the following suggestion.

1. The review of related work is not sufficiently thorough and not sufficiently specific. There are too few examples in the introduction. It is suggested to add more state-of-the-art research work from predecessors.

Answer: Based on the suggestion of the reviewer we have added more related works. By broadening the literature review, we believe to provide a more concrete context for our research. Please see the second paragraph of the section ``Introduction” (page no. 02) of the revised version of the manuscript.

2. The authors should ensure that the English language in this manuscript meets our high standards, is grammatically correct, and conforms to professional standards of courtesy and expression.

Answer: We have carefully reviewed and revised the manuscript to ensure that the English language meets high standards of clarity, grammatical correctness, and professionalism. We have also ensured that the writing conforms to standards of courtesy and expression.

3. The computational details are not mentioned. Which software, program languages, libraries, etc., were used to build this approach? Is it possible to publish code to use this architecture and reproduce the results? 

Answer: In our study, we employed a combination of software tools and programming libraries to implement the proposed approach, as outlined below: RStudio was primarily used for statistical analysis, data visualization, and multi-label classification tasks. We used Gephi for network visualization and Inkscape for figure refinement. Several R packages were utilized, including Csbl.go for the semantic similarity matrix, Utiml for multi-label classification, and factoextra for determining optimal clusters. Plotting was done with reshape, ggplot2, and ComplexHeatmap packages. We have utilized the ‘makeMultilabelClassifierChainsWrapper’ function of mlr R package for performing the classification. Additionally, we are making the code publicly available to enable other researchers to use the architecture and reproduce the results. The code can be accessed at the GitHub repository (https://github.com/Syed-Alberuni/NMF-for-Identification-of-Human-Disease-Associated-Gene-Module/upload/main).

4. The motivation part can be elaborated in the abstract, in the introduction, and the Discussion, if possible, which will enable the authors to improve the academic merit of their work.

Answer: We have elaborated on the motivation for our study in the abstract, introduction, and discussion sections. Additionally, the abstract has been updated based on feedback from other reviewers to better highlight the significance of our research and its contributions.

5. The authors only mentioned the results in the Discussion section, while there is no in-depth discussion.

Answer: We have expanded the Discussion section to include a more in-depth analysis of the results. This includes interpreting the findings in the context of existing literature, discussing their implications, and highlighting how they advance current knowledge in the field. We also address the potential limitations of the study and suggest directions for future research in the conclusions section.

Please see the updated (marked) subsections of the Results and Discussion section and the Conclusion section.

6. The figures and Tables quality needs to be improved.

Answer: Based on the suggestion of the reviewer we are now providing high-resolution images for all figures. Please note that we have made some minor changes to the figure.

7. Some technical aspects and essential insights of the proposed method are not described in detail.

Answer: The updated Method section provides an expanded explanation of the Non-Negative Matrix Factorization (NMF) technique used to integrate biological data sources, detailing its mathematical formulation and optimization process for identifying meta-modules. Additional details are included on the multi-label classification strategy, specifically the binary-relevance method, the classifiers evaluated, performance metrics such as Hamming loss, and the rationale for

selecting the K-nearest neighbor (KNN) classifier. Insights into the biological significance of the identified meta-modules are added, highlighting their role in understanding disease mechanisms and implications for precision medicine. The revisions also clarify the assumptions underlying the use of NMF and multi-label classification and discuss potential limitations of the method.

---

## [Decision Letter · Decision Letter 1]

22 Nov 2024

Integration of Biological Data via NMF for Identification of Human Disease-Associated Gene Modules through Multi-label Classification.

PONE-D-24-22050R1

Dear Dr. ALBERUNI,

We’re pleased to inform you that your manuscript has been judged scientifically suitable for publication and will be formally accepted for publication once it meets all outstanding technical requirements.

Kind regards,

Hilary A. Coller

Academic Editor

PLOS ONE

Additional Editor Comments (optional):

Reviewers' comments:

Reviewer's Responses to Questions

**Comments to the Author**

1. If the authors have adequately addressed your comments raised in a previous round of review and you feel that this manuscript is now acceptable for publication, you may indicate that here to bypass the “Comments to the Author” section, enter your conflict of interest statement in the “Confidential to Editor” section, and submit your "Accept" recommendation.

Reviewer #2: All comments have been addressed

Reviewer #3: (No Response)

2. Is the manuscript technically sound, and do the data support the conclusions?

Reviewer #2: Yes

Reviewer #3: (No Response)

3. Has the statistical analysis been performed appropriately and rigorously? 

Reviewer #2: Yes

Reviewer #3: (No Response)

4. Have the authors made all data underlying the findings in their manuscript fully available?

Reviewer #2: Yes

Reviewer #3: (No Response)

5. Is the manuscript presented in an intelligible fashion and written in standard English?

Reviewer #2: Yes

Reviewer #3: (No Response)

6. Review Comments to the Author

Reviewer #2: After a thorough review of the manuscript, I am pleased to recommend it for acceptance. The manuscript is well-prepared, clearly articulated, and meets the publication standards in terms of research integrity and ethical compliance. The manuscript contributes valuably to its field and should be shared with the academic community without further revisions.

Reviewer #3: According to the response letter, the paper has been revised well according to the previous reviewers, and the current version of the manuscript is acceptable for publication.

7. PLOS authors have the option to publish the peer review history of their article (what does this mean?). If published, this will include your full peer review and any attached files.

Reviewer #2: No

Reviewer #3: No

---

## [Editor Report · Acceptance letter]

2 Dec 2024

PONE-D-24-22050R1 

PLOS ONE

Dear Dr. Alberuni, 

I'm pleased to inform you that your manuscript has been deemed suitable for publication in PLOS ONE. Congratulations! Your manuscript is now being handed over to our production team.

Kind regards, 

on behalf of

Dr. Hilary A. Coller 

Academic Editor

PLOS ONE